# Quantification of Procedure Time and Infant Distress Produced (as Crying) When Percutaneous Achilles Tenotomy Is Performed under Topical Local Anaesthesia: A Preliminary Study

**DOI:** 10.3390/ijerph192113842

**Published:** 2022-10-25

**Authors:** Marta Vinyals Rodriguez, Anna Ey Batlle, Iolanda Jordan, Paula Míguez González

**Affiliations:** 1Equipo Ponseti Dra. Anna Ey, Clínica Diagonal, 08950 Barcelona, Spain; 2Hospital Sant Joan de Déu, 08950 Barcelona, Spain

**Keywords:** percutaneous Achilles tenotomy, local anaesthesia, ponseti method, crying, pain, infant

## Abstract

Introduction: Percutaneous tenotomy of the Achilles tendon is part of the clubfoot management procedure known as the Ponseti method and is necessary for most infants requiring this treatment. However, the need to apply general anaesthesia or sedation during this procedure remains controversial. To our knowledge, no previous studies have been conducted to quantify infant distress, expressed as crying, when tenotomy is performed under local anaesthesia. Material and Methods: This clinical, prospective, cross-sectional, and observational study was composed of infants subjected to percutaneous Achilles tenotomy with local anaesthesia at an outpatient clinic. The degree of distress was measured using two smartphone apps (voice recorder and timer) in two iPhones, with each apparatus placed one meter from the baby. The following parameters were determined: procedure duration, crying duration, average crying intensity and maximum crying intensity. In addition, the following data were obtained: age, complications (if any) and the caregiver’s satisfaction with the process. Results: Among the 85 infants submitted to percutaneous tenotomy, the mean age was 1.95 (+/−1.632) months (ranging from 0 to 7 months), the mean duration of the procedure was 8.134 (+/−5.97) seconds, (range 2.1 to 33.5 s), the infants’ mean crying intensity was 88.99 dB and the maximum crying intensity was 96.56 dB. No vascular or anaesthetic-related complications were recorded. 96% of the caregivers were absolutely satisfied with the process. Conclusions: Percutaneous Achilles tenotomy performed under local anaesthesia can safely be performed at the outpatient clinic. The procedure is fast and the crying time and intensity (mean values: 84 s and 89 dB, respectively) are minimal and tolerable. Knowledge of these parameters provides more accurate knowledge about the procedure. The caregivers consulted were absolutely satisfied with the tenotomy performed under local anaesthesia. In future studies, these parameters can be used for comparison with related surgical approaches.

## 1. Introduction

In the treatment of clubfoot, the Ponseti method consists of a progressive correction with plaster casts. According to previous research, in 95% of cases this approach includes percutaneous Achilles tenotomy [1,2,3,4,5].

In 1996, Dr. Ignacio V. Ponseti reported a longitudinal prospective observational study according to which clubfoot was successfully corrected by a procedure including percutaneous Achilles tenotomy in 98.9% of the cases considered, with no residual pain [2].

In the Ponseti method, percutaneous Achilles tenotomy should be performed when 60–70° of abduction is obtained by the casts, and the heel is in valgus. If the tenotomy is not performed and the forced dorsiflexion movement is performed with the casts in place, a rocker bottom deformity or a flattening of the talus may be provoked [6].

Although this technique is minimally invasive, with few complications [7,8], and requires only basic surgical material, the need to perform it under general anaesthesia or after sedation remains a matter of controversy. In Ponseti’s original proposal, the tenotomy is performed with local anaesthesia and at the outpatient clinic; this approach has been confirmed in later reports by the same author [2,8,9].

To our knowledge, the differences between performing a tenotomy under local or general anaesthesia have not been clearly measured, but many references have been made to the infant’s suffering if the surgery were performed under local anaesthesia, a hypothesis that would justify the application of sedation or general anaesthesia.

In the present study, we analyse the viability of performing percutaneous Achilles tenotomy as an outpatient procedure under local anaesthesia. This analysis is based on an initial quantification of the suffering involved and of related variables.

To our knowledge, only one previous study has been conducted to quantify the time required to apply the Ponseti cast, and none have been undertaken to evaluate the infant’s suffering in terms of the crying produced or the time elapsed during the procedure.

Aim: Our study aim is to quantify the pain and discomfort generated by percutaneous Achilles tenotomy as an outpatient activity performed under local anaesthesia.

## 2. Material and Method

This clinical study is prospective, observational, and cross-sectional. The participants were consecutive patients with clubfoot who had undergone percutaneous tenotomy of the Achilles tendon after the application of a corrective cast following the Ponseti protocol. In every case, the procedure was performed at the outpatient clinic in the presence of the family and always by the same professional, who was an expert in this method.

Inclusion criteria: Patients with congenital clubfoot, treated by the Ponseti method and whose age corresponds to the lactation stage (0 to 24 months).Exclusion criteria: Patients with neurological involvement and an associated pathology.

The following demographics variables were analysed: 85 participants (53 male, 32 female) were studied. Thirty-seven of the participants’ clubfeet were bilateral and 48 were unilateral. The tenotomy was practiced in all patients (112 clubfeet, across 85 participants).

The mean of number of casts done before the tenotomy was 2,96 (range). The initial Pirani score mean was 5,95 (range).

We have performed the tenotomy in all patients with an opthalmic blade. It offers easier handling, precision, and control than a needle, and the incision is smaller than that of a 15 or 11 scalpel blade. Although all instruments are valid, it depends on the comfort of the professional to perform the percutaneous tenotomy procedure.

The calming effect used in this study was commercial milk in 30% of cases, breastfeeding in 13.34%, sucrose in 10%, white noise/music/cartoons in 10%, and lastly, a pacifier in 36.66 % of the total. Sucrose is a very interesting option as a calming effect on the baby after tenotomy. All options are valid. It depends on each baby and the parents.

The following variables were quantified: age, duration of procedure, mean intensity of crying, maximum intensity of crying, and crying time. In every case, the mean (Χ), standard deviation (SD), and minimum and maximum values were calculated.

In addition, the following qualitative variables were obtained: any complications arising from the procedure and the results of the caregiver satisfaction survey.

After receiving approval for the study from the Hospital’s ethics committee, the caregivers were given written information on the study’s goals, and their informed consent was obtained. Data collection then took place, including the patient’s clinical history and the study variables detailed above. This study was conducted in full compliance with the provisions of the Declaration of Helsinki. Data analysis was carried out using Excel and SPSS (v.15.0.1).

Previous tenotomy Ponseti cast (60–70° abduction cast) removal.Application of topical local anaesthesia and occlusive dressing. We use a lidocaine and prilocaine preparation (cream) and we apply it with an occlusive dressing during 45 min after retiring the cast with 60–70° of abduction.Asepsis of the area.A 1-cm medial to lateral incision made, proximal to the calcaneal insertion, using an ophthalmic bladeComplete section of the tendon, applying pressure to the area.Application of adhesive skin closures.Application of plaster cast, maintaining a dorsiflexion of 15–20° and the 60–70° abduction achieved with the previous cast, for 15 days.Voice recording (using Voice Memo App for iPhone X), placing the smartphone at a distance of 1 m from the infant.The recording starts when the scalpel is first applied and ends when the click of the complete tenotomy is heard and evidenced by the descent of the calcaneus.Crying intensity is measured using the Decibel 10 App (for iPhone X).The recording begins three minutes before the start of the procedure and continues until the crying stops.

## 3. Results

Data were analysed for 85 patients who underwent the percutaneous tenotomy procedure. These patients had a mean age of 1.95 (+/−1.632) months, ranging from 0 to 7 months. Of this population, 25% were less than one month of age and 75% were less than or equal to three months of age. The mean duration of the tenotomy procedure was 8.134 (+/−5.97) seconds, ranging from 2.1 to 33.5 s. In 75% of the cases, the duration was less than 10.18 s.

Duration of the surgical procedure (Figure 1)

Mean duration: 8.134 s.Standard deviation: 5.9798.Maximum value: 33.5 s; minimum value: 2.1 s.For 25% of patients, the duration was ≤4.115 s.For 50% of patients, the duration was ≤6.450 s.For 75% of patients, the duration was ≤10.185 s.

The mean initial crying time after tenotomy was 84.247 (±46.73) seconds, ranging from 13.7 to 267.4 s. In 75% of cases, the time was ≤92.58 s (Figure 2).

Mean duration: 84.247 s.Standard deviation: 46.7309.Maximum value: 267.4 s; minimum value: 13.7 s.For 25% of patients, the duration was ≤58.55 s.For 50% of patients, the duration was ≤75.12 s.For 75% of patients, the duration was ≤92.58 s.

The intensity of crying was measured by dividing the mean value by the maximum, in decibels (Figure 3 and Figure 4).

Intensity of crying:Mean intensity: 88.999 dB.Standard deviation: 8.6189.Maximum value: 107.3 dB; minimum value: 67.6 dB.For 25% of patients, the mean intensity of crying was ≤81.9 dB.For 50% of patients, the mean intensity of crying was ≤89.1 dB.For 75% of patients, the mean intensity of crying was ≤96.05 dB.

Maximum intensity of crying:Mean intensity: 96.568 dB.Standard deviation: 8.1926.Maximum value: 116.2 dB; minimum value: 79.3 dB.For 25% of patients, the maximum intensity of crying was ≤90.15 dB.For 50% of patients, the maximum intensity of crying was ≤96.3 dB.For 75% of patients, the maximum intensity of crying was ≤103.35 dB.

No complications due to bleeding or the application of topical anaesthesia were reported. Regarding caregivers’ satisfaction with the tenotomy, according to the replies given to a five-item survey in this respect, 96% were “absolutely satisfied” with the procedure.

## 4. Discussion

Most reports in this area describe the performance of the percutaneous Achilles tenotomy with sedation or general anaesthesia, and several protocols have been proposed [10,11]. This use of anaesthesia is recommended in order to avoid suffering for the baby and to better control the limb, thus enhancing the safety of the procedure. In contrast to previous research [8], our study revealed no complications in this respect.

We identify a research gap concerning the measurement of infant suffering before and during the percutaneous tenotomy with local anaesthesia in the outpatient clinic. Thus, regarding the Ponseti method, only one study has measured the time involved, and this analysis only concerned the application of the cast, not the tenotomy procedure.

Various scales can be used to assess infant stress, such as the Premature Infant Pain Profile (Stevens, 1996) and the Neonatal Infant Pain Scale (Laurence 1993). However, all seek to measure stress at the moment when discomfort increases in the hospital environment, or in relation to specific behavioural and physiological parameters, for example during admission to intensive care.

In addition, analyses have been made of short, painful procedures that could be equated to tenotomy, such as heel puncture, and which use crying time as a proxy for pain [12]. We decided to use this parameter as well, since the procedure considered is known to provoke stress. However, our specific aim is to assess how long and how much the baby suffers in an outpatient environment, accompanied by family caregivers, where there may not be facilities for determining other variables such as oxygen saturation, heart rate or respiratory rate. In the present study, we propose a measurement approach that is repeatable anywhere the Ponseti method is performed.

The use of local or general anaesthesia in performing percutaneous Achilles tenotomy is controversial, since it has been argued that general anaesthesia in infants aged under three months is associated with a greater risk of complications, some of which can be severe, such as apnoea [13,14]. For this reason we consider it important to consider whether sedation or general anaesthesia before this relatively minor procedure is really necessary. Moreover, the use of general anaesthesia in infants has been related to the appearance of later problems such as attention disorders in childhood, which has been associated with the anaesthetic drugs used [15,16]. These consequences have also been examined in conjunction with the possible repercussions of future suffering [12], such as changes in sensory and somatosensory processes in response to future painful stimuli, neurosensory development disorders and alterations in emotional and learning behaviour [16].

Although the use of an operating room is recommended to perform an open percutaneous tenotomy, as described in the Ponseti method [17], it can also be performed in the outpatient clinic, under local anaesthesia. This approach offers the following advantages: the financial cost is lower; the infant need not be fasting, and the surgery can be performed in the presence of family members. In contrast, hospital admission usually involves fasting and separation of the infant from caregivers (situations that are both associated with crying).

In this preliminary study, we establish values quantifying infant suffering during a surgical procedure, measured according to the duration and intensity of crying, in a way that is simple, accessible (via a smartphone app), and reproducible. With these parameters, comparative studies can be carried out with the same protocol, whether concerning percutaneous tenotomy performed in other settings, or other more or less invasive procedures such as heel puncture, vaccination, the insertion of earrings or the cutting of a frenulum, which most professionals assume should be performed without anaesthesia.

We believe the data reported in this paper will allow clinicians to better inform the families of patients with clubfoot about this procedure, thus enhancing their peace of mind and decision-making capacity. Nevertheless, this study is merely preliminary and is intended to lay the foundations for later studies to generate a broad-based protocol for measuring infant crying in comparable contexts.

## 5. Conclusions

Percutaneous Achilles tenotomy using topical local anaesthesia can safely be performed in the outpatient clinic. The procedure is fast. The mean duration (84 s) and intensity (mean 89 dB) of crying is minimal and tolerable. This specific information about the procedure will be reassuring for caregivers, who state they are “absolutely satisfied” with the tenotomy performed as described in both unilateral and bilateral instances. In future studies, the parameters we have considered can be used to compare the outcomes from tenotomy with those from other procedures.

## Figures and Tables

**Figure 1 ijerph-19-13842-f001:**
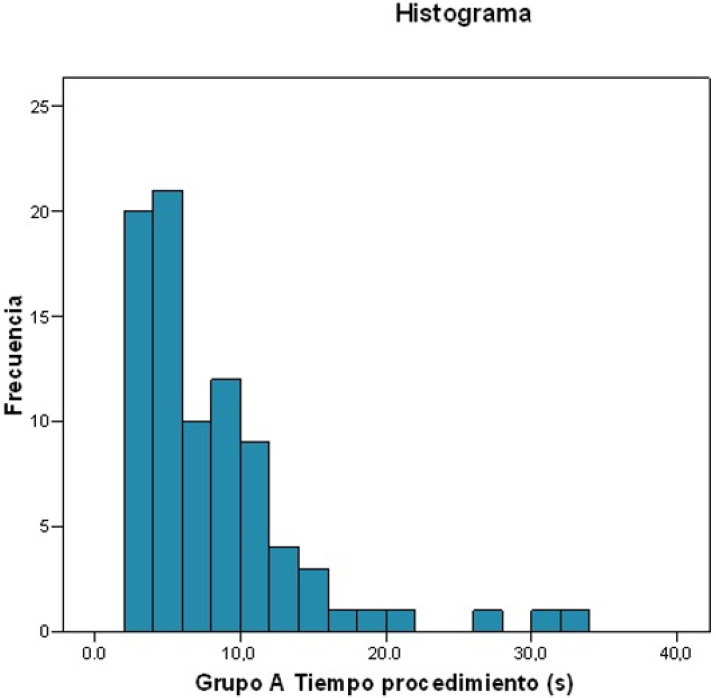
Histogram: Frequency of the duration of percutaneous Achilles tenotomy.

**Figure 2 ijerph-19-13842-f002:**
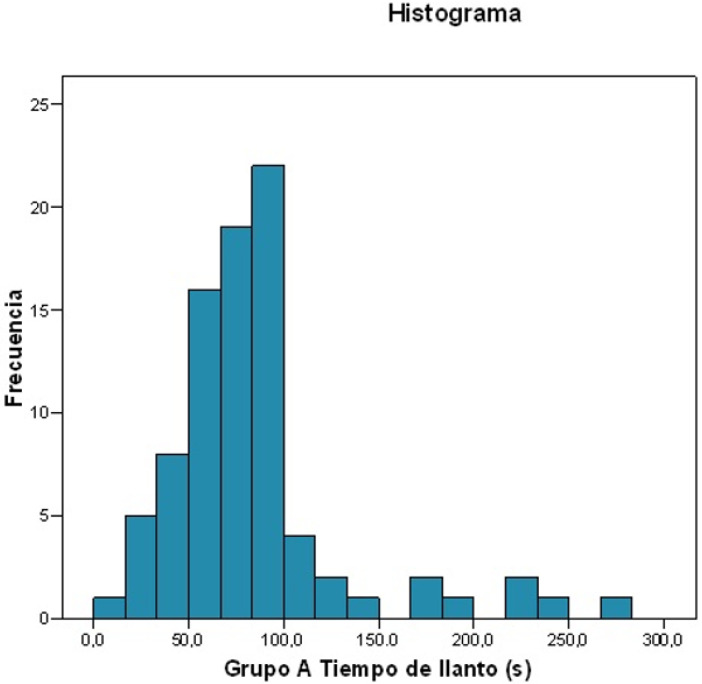
Histogram: Frequency of the duration crying time.

**Figure 3 ijerph-19-13842-f003:**
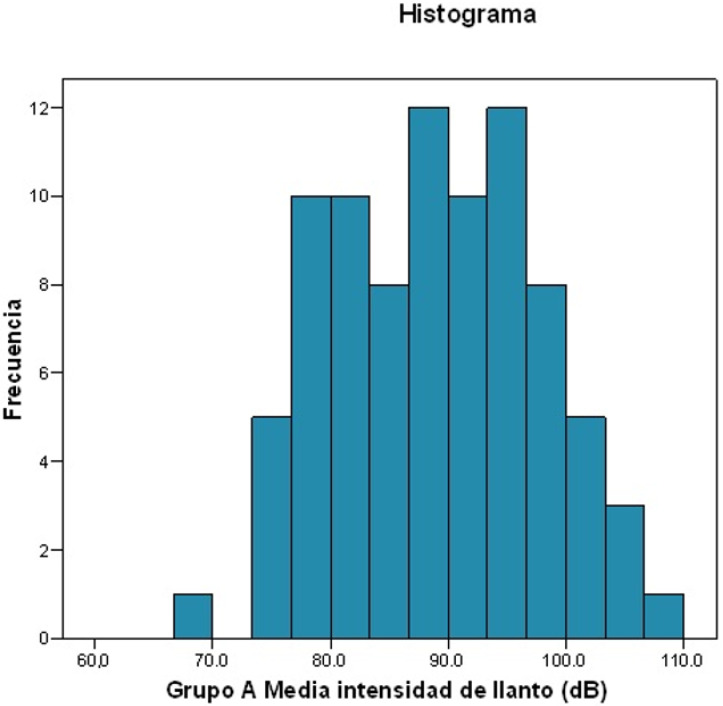
Frequency histogram about crying intensity.

**Figure 4 ijerph-19-13842-f004:**
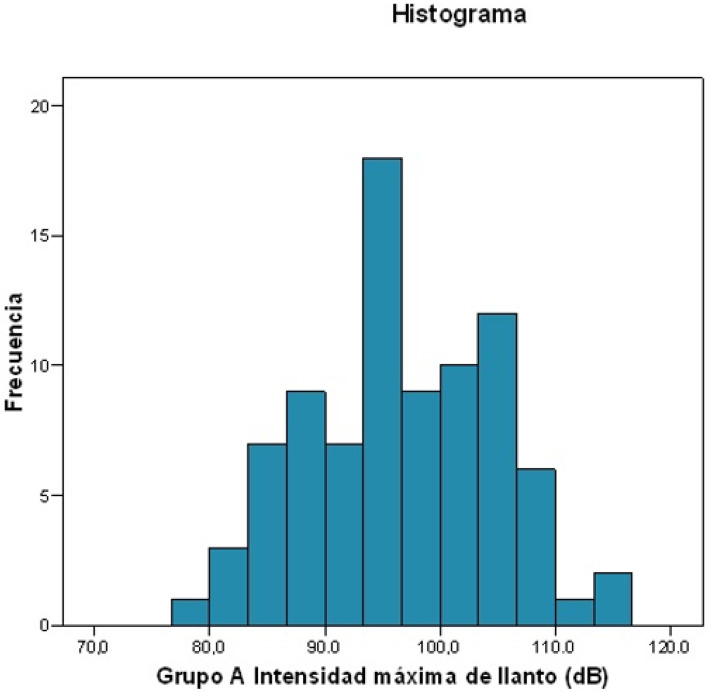
Frequency histogram about maximum crying intensity.

## Data Availability

Not applicable.

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
