# Peer review of "Quantification of Procedure Time and Infant Distress Produced (as Crying) When Percutaneous Achilles Tenotomy Is Performed under Topical Local Anaesthesia: A Preliminary Study"

_ijerph, 2022, doi:10.3390/ijerph192113842_

Round 1

Reviewer 1 Report

After reading and analyzing the article, I propose to reject it because:

- local anesthesia only with cream has a very good effect only on the skin, not on the profuse tissues.

- the risks and stress of the child seem to me higher for this intervention performed in the outpatient department, compared to the intervention performed in the surgery room, with sedation and analgesia.

- in addition, the asepsis conditions are higher in the surgery room, so the risks are lower.

From the point of view of the actual study, there are some biases:

- the level of intensity of crying is different from one child to another and depends on the pain threshold of each patient; it seems subjective to me and not objective.

- pain level should be measured by other methods (respiratory rate, heart rate, pulse).

- I'm not talking about the associated abnormalities, which are common in the case of the clubfoot and could influence the results.

The article is not well structured, the Limitation of the study is missing. Also, few comparisons with the results of other studies.

The opinion of the ethics commission of the hospital and the families consent are missing.

Author Response

Ponseti Method Protocol is described since 1950 and used in all over the world with 100% correction. In our team we are doing this protocol since 1997 , more that 1000 babies treated every year with this protocol with 100% correction obtained. If i accept your suggestions i will be failing my principles and principles of this Method. In this 25 years we can demonstrate that we are absolutelly disagree with your suggestions

Reviewer 2 Report

The authors perform a study in 85 infants during the percutaneous tenotomy, aiming to solve the controversy on the need of sedation instead of the current used local anesthesia. 

I need to raise some questions:

1. What were the comparison values and studies for stating in lines 25-26:  the crying time and intensity (mean values: 84 seconds 25 and 89 dB, respectively) are minimal and tolerable. More references on other studies that perform assessment of infants distress must be included in the text. 

2. What is the number of approval from the Ethical Committee and the registration to clinicaltrials.gov/ similar registry?

3. What is in the 96-106 lines?

4. More details should be provided for the comparisons to references 8 (what are the other complications), 12 (what crying time did they considered a proxy for pain). More references on similar studies must be provided.

Small English corrections should also be provided: repetition of "perfprmed" in the same sentence (line 41).

Author Response

Dear reviewer,

First of all, thank you for your help and comments.

  1. There are no similar papers published with this comparasions. My second paper ready to publish is a comparison of this results with another procedure. I think it will be also a good and useful contribution.
  2. I attach in this repsonse the aprovation of ethical commitee
  3. I cannot find these lines, could you please mark which lines do you mean?
  4. Same as first answer, there are no similar papers published to compare
  5. Could you please mark where is this line? because ive re-read 3 times the article and i really can not find it. I am sorry

Thank you so much for your help and appreciations

Round 2

Reviewer 1 Report

Indeed, the Ponseti technique is widely used today. It can be performed on an outpatient clinic until the percutaneous Achilles tenotomy. I still believe that the asepsis conditions are superior in the operating room, so the risks are lower.